# Ultra-dense dislocations stabilized in high entropy oxide ceramics

Yi Han [1,4], Xiangyang Liu[1,4], Qiqi Zhang[2], Muzhang Huang[1], Yi Li [1], Wei Pan [1,5✉], Peng-an Zong[1], Lieyang Li[1], Zesheng Yang[1], Yingjie Feng[1], Peng Zhang [1,3,5✉] & Chunlei Wan [1,5✉]

Dislocations are commonly present and important in metals but their effects have not been fully recognized in oxide ceramics. The large strain energy raised by the rigid ionic/covalent bonding in oxide ceramics leads to dislocations with low density ($\sim10^6\,mm^{-2}$), thermodynamic instability and spatial inhomogeneity. In this paper, we report ultrahigh density ($\sim10^9\,mm^{-2}$) of edge dislocations that are uniformly distributed in oxide ceramics with large compositional complexity. We demonstrate the dislocations are progressively and thermodynamically stabilized with increasing complexity of the composition, in which the entropy gain can compensate the strain energy of dislocations. We also find cracks are deflected and bridged with $\sim70\%$ enhancement of fracture toughness in the pyrochlore ceramics with multiple valence cations, due to the interaction with enlarged strain field around the immobile dislocations. This research provides a controllable approach to establish ultra-dense dislocations in oxide ceramics, which may open up another dimension to tune their properties.

[1] State Key Laboratory of New Ceramics and Fine Processing, School of Materials Science and Engineering, Tsinghua University, 100084 Beijing, China. [2] National Center for Electron Microscopy in Beijing, 100084 Beijing, China. [3] Institute of Welding and Surface Engineering Technology, Faculty of Materials and Manufacturing, Beijing University of Technology, 100124 Beijing, China. [4] These authors contributed equally: Yi Han, Xiangyang Liu. [5] These authors jointly supervised this work: Wei Pan, Peng Zhang, Chunlei Wan. ✉email: panw@mail.tsinghua.edu.cn; peng.zhang@bjut.edu.cn; wancl@mail.tsinghua.edu.cn

islocations, a line imperfection in crystals formed by local displacements of atoms from the periodic arrangement, are widely present in metals, alloys, and intermetallic compounds (borides, carbides) and play important roles in their vast mechanical properties and functional properties[1–7]. The presence of high-density dislocations in metals and metallic compounds is mainly due to the non-directional and non-specific nature of metallic bonds that can relax and accommodate the strain caused by dislocations[8,9]. In contrast, dislocations are rather uncommon and much less important in oxide ceramics until now. The major reason is that oxide ceramics are composed of rigid and stiff ionic/covalent bonds and can hardly tolerate the large strain energy associated with dislocations, which is quite different from metals or metallic compounds[10,11]. There have been many reports to induce dislocations in oxide ceramics, such as high-temperature deformation[12–15]. However, these dislocations always show low density ($10^6$ mm$^{-2}$), which is two orders of magnitude lower than that in metals or alloys ($10^8$ mm$^{-2}$)[14,16]. Meanwhile, the dislocations are always thermodynamically unstable, and annealing at high temperatures would cause the annihilation of the dislocation[17]. There are also efforts to induce dislocations around interfaces between two dissimilar materials, but the dislocations are localized in the region close to the interface, with a little influence on the bulk material properties[18,19]. Therefore, the effect of dislocations on the properties of oxide ceramics has not been fully recognized yet.

Particularly, dislocations may ameliorate the brittleness of oxide ceramics, which has long been a major concern that prevents their wider applications. It has recently been identified that high-density dislocations can enable local plasticity on the surface of a ceramic crystal and may become an additional toughening mechanism in bulk ceramics[20]. It also shows that the strain field around the dislocations can also contribute to the enhancement of fracture toughness through the stress shielding effect[21]. However, both routes require a high density of dislocations to the level of $10^8$ mm$^{-2}$ in bulk ceramics[20,21], which is beyond the reach of the available techniques.

Meanwhile, high-entropy ceramics have attracted rapidly increasing interests in wide-ranging fields, which bring opportunities in tuning the microstructure and properties of oxide ceramics[22–33]. Although varieties of interesting properties have been intensively reported for high-entropy ceramics, an in-depth understanding of their microstructure has not yet come up. Until now, it is widely considered that the constitutional elements are randomly distributed on each crystallographic site, showing compositional heterogeneity at the atomic scale[34]. More insights into the microstructure of the high-entropy oxide ceramics are strongly required to further understand their abnormal properties and accelerate the exploration of unprecedented properties and applications.

In this work, we report a universal microstructural feature that is hidden in high-entropy oxide ceramics. There are actually ultrahigh-density dislocations (~$10^9$ mm$^{-2}$) in high-entropy oxide ceramics. These dislocations are thermodynamically stabilized because the configuration-entropy gain can compensate for the large strain caused by the rigid ionic/covalent bonds in oxide ceramics. We further demonstrate the density of dislocations can even be further tuned by the entropy gain with the increase of composition complexity. It should be mentioned that dislocations are always seen in both simple and high-entropy intermetallic compounds, such as borides and carbides, but these materials contain metallic bonds to accommodate the strain of dislocations, showing a different scenario with the oxide ceramics[35]. Meanwhile, we show that the ultra-dense dislocations show an extra toughening effect through interaction with the cracks. The strain field of dislocations in the high-entropy oxide ceramics can be enlarged by using multiple valent cations, which can induce higher chances of deflection and bridging of cracks with a large improvement in mechanical toughness. This work provides a controllable approach to establish dislocations with ultrahigh density in oxide ceramics, which brings opportunities in tuning mechanical, chemical[36], electrical[37], and transport properties[38], with potential technological applications, such as thermal barrier coatings[39,40], solid oxide fuel cells[41], photovoltaics[42], ferroelectics[43], dielectrics[44], and thermoelectrics[45,46].

## Results

**Ultrahigh-density dislocations discovered in high-entropy fluorite oxide.** The atomic microstructure of rare-earth zirconate ($Gd_2Zr_2O_7$) and high-entropy fluorite oxide ($(Sm_{0.2}Gd_{0.2}Dy_{0.2}Er_{0.2}Yb_{0.2})_2Zr_2O_7$, HEFO) is observed by the high-resolution transmission electron microscope (HRTEM) along the [110] zone axis. The atomic images are treated by the Fourier analysis and geometric phase analysis (GPA) to obtain detailed structure and strain information. In Fig. 1a, the atoms of $Gd_2Zr_2O_7$ are neatly arranged, without any lattice distortion or imperfections. Fourier analysis is taken to clearly show the lattice feature of $Gd_2Zr_2O_7$, which consists of a fast Fourier transform (FFT) of the atomic image, ring-shape masking with specific frequency in reciprocal space (Supplementary Fig. 3), and an inverse fast Fourier transform (IFFT) (Fig. 1b, d). Figure 1b is the corresponding IFFT image of Fig. 1a with a normal crystallographic direction of the lattice planes (111). The uniform fringe patterns in the IFFT image indicate that there are no lattice distortion and imperfections in $Gd_2Zr_2O_7$, which is consistent with the atomic image in HRTEM. With the increase of the composition complexity, ultra-dense edge dislocations are observed in the atomic image of HEFO in Fig. 1c. Edge dislocations in HEFO can be observed clearly in the yellow square area in Fig. 1c. A single edge dislocation (correspond to the upper left edge-dislocation in Fig. 1c) with Burgers vector of 1/2[111] is enlarged and shown in Fig. 1e. The corresponding IFFT image for Fig. 1c is displayed in Fig. 1d, which is different from the IFFT image of $Gd_2Zr_2O_7$. Firstly, instead of the uniform fringe patterns, edge dislocations with the same Burgers vector can be observed in Fig. 1d, which corresponds well with the atomic image in real space. One edge-dislocation (marked by the red frame in Fig. 1c, d) is not found in Fig. 1d due to the different Burgers vectors that cannot be observed along this crystallographic direction. There are always two types of dislocations with different Burgers vectors. Via drawing the Burgers circuit around the dislocation, the Burgers vectors are determined as 1/2[111] and 1/2 [200]. The number of dislocations with Burgers vectors of 1/2[111] is more than the number of dislocations with other Burger vectors. That is due to the smaller interplanar distance of {111} and less strain energy. Secondly, lattice distortion is also found in Fig. 1d in the vicinity of the dislocation area. Indeed, there are still some "dislocation-like" fringe patterns in the dark area in Fig. 1d, but they cannot be identified as dislocations as no corresponding Burgers circuit can be drawn in Fig. 1c. The strain distribution of Fig. 1c is calculated by GPA and the corresponding strain mapping is shown in Fig. 1f. For an isolated single edge dislocation, it consists of a symmetrical compression-tension strain pair. To calculate the dislocation density, we count the number of dislocations in three independent TEM images for each composition. All the HRTEM images are taken at random positions of samples to reduce the randomness error. (The detailed method to confirm a dislocation is introduced in the Supplementary materials). The dislocation density is then calculated to be the average number of dislocations per unit area (mm$^{-2}$). The dislocation density of HEFO is around $10^9$ mm$^{-2}$, which is much higher than the conventional oxide ceramics (~$10^6$ mm$^{-2}$), and even higher than that in some metals or alloys (~$10^8$ mm$^{-2}$). The high dislocation density of

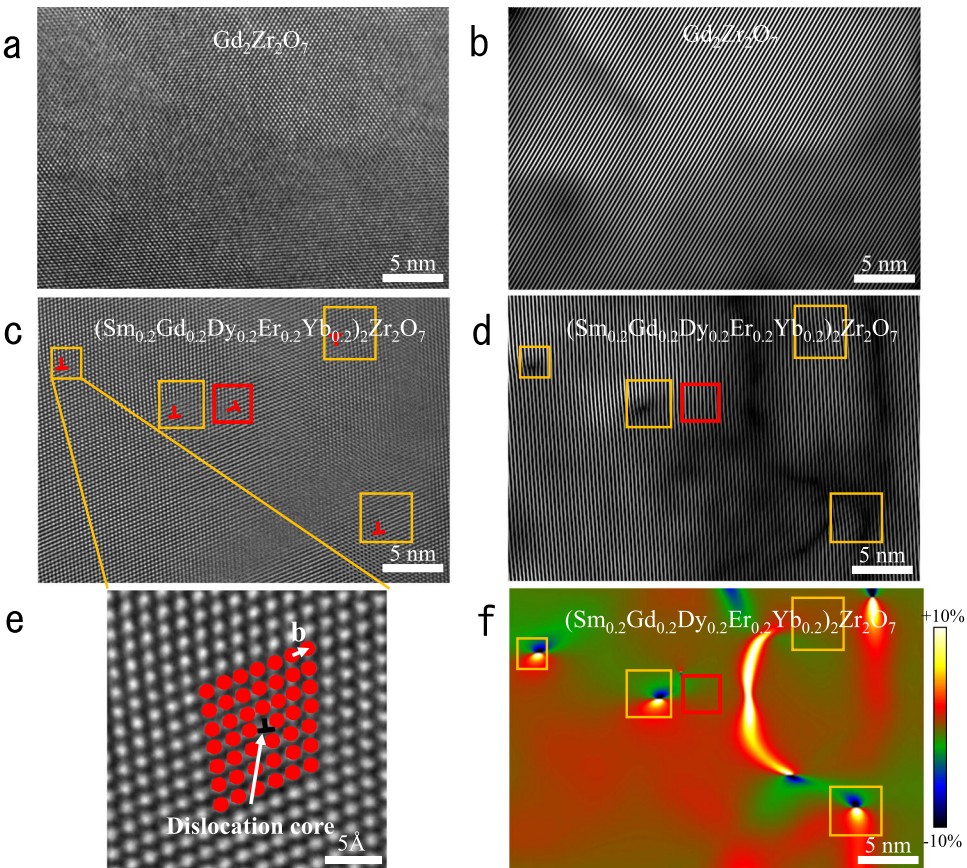

**Fig. 1 High-resolution characterization of Gd$_2$Zr$_2$O$_7$ and high-entropy fluorite oxide (HEFO, (Sm$_{0.2}$Gd$_{0.2}$Dy$_{0.2}$Er$_{0.2}$Yb$_{0.2}$)$_2$Zr$_2$O$_7$) with edge dislocations and strain distribution. a** High-resolution transmission electron microscope (HRTEM) graph of Gd$_2$Zr$_2$O$_7$; **b** The inverse fast Fourier transform (IFFT) filtered image of Gd$_2$Zr$_2$O$_7$ shows perfect periodicity without dislocations; **c** HRTEM graph of high-entropy fluorite oxide; **d** The inverse fast Fourier transform (FFT) filtered image of high-entropy fluorite oxides; **e** HRTEM graph of single edge dislocation; **f** Geometric Phase Analysis (GPA) mapping of **a** calculated by Digital Micrograph, showing an overall strain field at nanoscale. The color scale represents change in strain intensity from −10% (compressive) to 10% (tensile). The yellow and red squares mark the area around the dislocations.

high-entropy oxide ceramics may cause complex strain distribution that could have some positive effects on the properties.

**Formation mechanism of ultra-dense dislocations.** In order to elucidate the formation mechanism of ultra-dense dislocations in HE oxide ceramics, molecular dynamic simulation is performed to study the strain energy of dislocations in HEFO. As shown in Fig. 2b, one edge-dislocation is successfully introduced in the model. About four cation layers are distorted intensively around the edge-dislocation core, which is consistent with the TEM image displayed in Supplementary Fig. 7. Moreover, the cation lattice away from the edge dislocation is linearly arranged, whereas the anion lattice is still strongly distorted. The cations in the HEFO model exhibits a typical fluorite structure, and they are randomly distributed in the cation sublattice which satisfies the concept of high-entropy materials. Moreover, as shown in Supplementary Fig. 13, the positions of the two dislocations are slightly changed to accommodate themselves to the lattice strain in order to minimize the total energy.

To demonstrate the thermodynamic stability of the ultrahigh dislocations in the high-entropy oxides, the free energy of the system is estimated as follows:

$$\triangle F = \triangle H - T\triangle S \tag{1}$$

where $\triangle H$ is the enthalpy increase of the high-entropy oxides containing high-density dislocations, which is calculated from the energy difference between high-entropy oxide and the five individual

components (Sm, Gd, Dy, Er, and Yb) with the same structure (fluorite structure)[47]. In fact, it has two contributions. One is the mixing enthalpy of the five individual components $\Delta H_{mix}$ and the other is the mean strain energy raised by the dislocation $< E_{strain} >$. It is found that $< E_{strain} >$ dominates the total enthalpy increase (>85%). The increase of entropy $\Delta S$ includes two contributions, the entropy increases due to the formation of a dislocation $\Delta S_{dislocation}$ and the mixing entropy of the atoms that comprise the dislocation $\Delta S_{mix}$. The contribution of dislocations to the entropy increase $\Delta S_{dislocation}$ can be determined via the definition of Shannon entropy[48,49].

$$\triangle S_{dislocation} = -k_B[p_i \ln p_i + (1 - p_i)\ln(1 - p_i)] \tag{2}$$

Where $k_B$ is the Boltzmann constant, $p_i$ is the possibility to detect the dislocation in the model which is proportional to the total number of cations in the dislocation line (~0.34%). $\Delta S_{mix}$ is the entropy difference from the contribution of ideal mixing. In our calculation, $\Delta S_{mix}$ is about 1.6 $nk_B$, and $n$ is the number of unit cells in the calculation that should cover all the strain area. As shown in Supplementary Fig. 12, the strain-existence zone can be covered by a cylinder (the center is set to be the edge-dislocation core) with a radius of ~2 nm which contains 252 unit cells. For the HE-Model, the $\Delta S_{dislocation}$ is 0.0236 $k_B$ per cation which is ~1.4% of the $\Delta S_{mix}$. As a result, the total entropy difference $\Delta S$ can be obtained by combining the $\Delta S_{dislocation}$ and $\Delta S_{mix}$. The enthalpy change of an edge dislocation and entropy difference is displayed in Fig. 2d. It shows the enthalpy change induced by edge dislocations can be

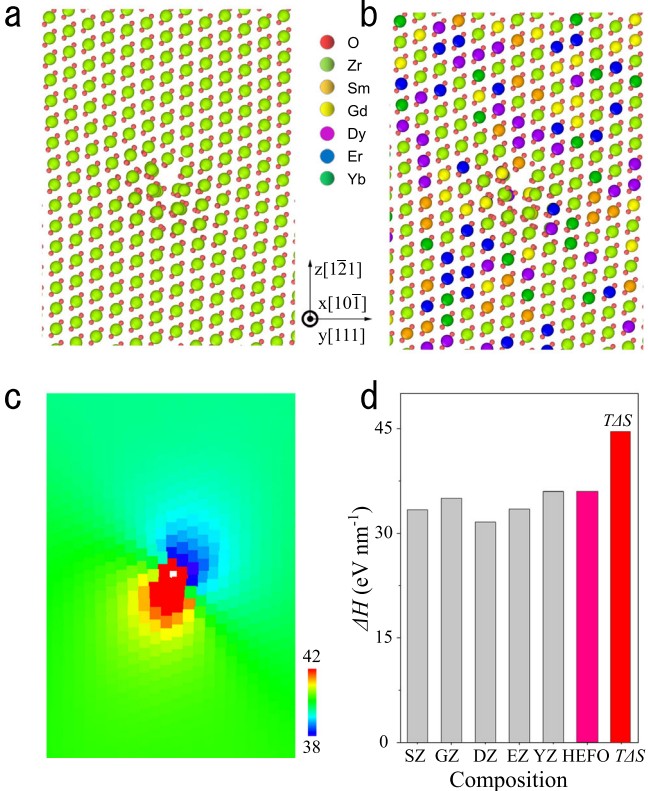

**Fig. 2 The edge-dislocation model created for MD simulations. a** Before doping with rare-earth ions; **b** After doping with rare-earth ions; **c** Voronoi atomic volumes distribution of the cations calculated by OVITO software (The color scale denotes the Voronoi volume of each cation (Å³)); **d** Enthalpy change of an edge dislocation compared with the value of "$T\Delta S$". (SZ, GZ, DZ, EZ, and YZ represent $Sm_2Zr_2O_7$, $Gd_2Zr_2O_7$, $Dy_2Zr_2O_7$, $Er_2Zr_2O_7$, and $Yb_2Zr_2O_7$ respectively.).

compensated by the configuration-entropy gain, resulting in thermodynamic stability. As a result, the high-entropy ceramics with ultrahigh dense dislocations have a much smaller Gibbs free energy compared with individual rare-earth zirconates. In other words, configurations with micro-scale strains are allowed in high-entropy oxide ceramics and the tolerance of strain energy increases with the increase of composition complexity and configuration entropy.

To further confirm the effect of composition complexity, the microstructure and dislocation density of the rare-earth zirconates are studied as a function of the number of the principal elements (N) and configuration entropy in Fig. 3a-d. $(Gd_{0.5}Er_{0.5})_2Zr_2O_7$ with three principal elements has a similar atomic image as $Gd_2Zr_2O_7$ (Fig. 1a), without any dislocations found in the HRTEM graph. With the increase of N, dislocations are observed in the atomic image of fluorite oxides with quaternary and quinary principal elements. Further increase N to six that is the high-entropy fluorite oxide $(Sm_{0.2}Gd_{0.2}Dy_{0.2}Er_{0.2}Yb_{0.2})_2Zr_2O_7$, in which many more dislocations are then observed in Fig. 3d. A well-fitted curve for the dislocation density of fluorite oxides with different N is shown in Fig. 3e. It can be clearly seen that the dislocation density dramatically increases to ~$10^9$ mm$^{-2}$ when N = 6, which is an order of magnitude higher than the other fluorite oxides with fewer principal elements. The results demonstrate that the dislocation density is highly sensitive to the composition complexity. To further confirm the underlying mechanism, the thermodynamic parameters are estimated, including the enthalpy change and the configuration entropy in Fig. 3f. It can be found the enthalpy change of dislocations in fluorite oxides is

almost insensitive to the number of principal elements, N. In contrast, with the increase of N, the entropy gain $T\Delta S$ gradually increases and exceeds the value of enthalpy change of dislocation after N = 4, leading to a minus free energy. Meanwhile, dislocations appear for the composition of N = 4 and the density increases dramatically. The composition complexity actually determines the configurational entropy of ceramics. These results demonstrate the vital role of configuration entropy of the complex ceramics on the stabilization of high-density dislocations. It also points out a route to manipulate the dislocation density in oxide ceramics through tuning the composition complexity.

To further demonstrate the universality of this strategy, we examine the dislocation densities of various high-entropy oxides ceramics with different structures, including fluorite $(Dy_{0.25}Er_{0.25}Y_{0.25}Yb_{0.25})_3NbO_7$, perovskite $Sr(Ti_{0.2}Y_{0.2}Zr_{0.2}Nb_{0.2}Sn_{0.2})O_3$, fergusonite $(Nd_{0.2}Sm_{0.2}Gd_{0.2}Er_{0.2}Yb_{0.2})NbO_4$, and pyrochlore $Ca_{1.2}Gd_{0.8}Zr_{0.8}Nb_{0.6}Ta_{0.6}O_7$. HRTEM images of these ceramics are shown in Fig. 4a-d, where the electron beam is parallel to the [110] zone axis. The dislocations verified by the same method are marked by orange rectangles. Ultra-dense edge dislocations, up to the order of $10^9$ mm$^{-2}$, are also observed in various high-entropy oxide ceramics without any special treatment. In addition, such dislocations are also observed in $(Ca_{0.25}Sr_{0.25}Ba_{0.25}RE_{0.25})TiO_3$ recently[46]. It demonstrates the applicability of composition complexity to regulate the dislocation density of oxide ceramics with different structures. The fitted curve calculated by the dislocation density and configurational entropy is shown in Fig. 4e. Dislocations almost do not exist and have little change in the low entropy part. Once the configurational entropy gain exceeds a certain value related to the strain energy of dislocations, the dislocation density increases rapidly. That is consistent with the simulated results in Fig. 3f. Therefore, configurational entropy becomes the main driving force of ultra-dense edge dislocations. Although oxide ceramics with different structures have similar configurational entropy, the dislocation density may be different. This is due to the different strain energy of dislocations of various crystal structures, which need different entropy gain to compensate. For example, both high-entropy fluorite oxides and perovskite oxides are face-centered cubic (FCC) structures. The FCC structure is a high symmetry structure with multiple slip systems, having lower dislocation strain energy and leading to higher dislocation density of high-entropy oxides. On the contrary, high-entropy fergusonite oxides have a low-symmetry monoclinic phase, partially owing to the presence of ferroelastic domains. Therefore, despite the high-entropy fergusonite oxide ceramics having the same configurational entropy, these ceramics possess fewer dislocations as compared to the high-entropy oxide ceramics with FCC structure.

**Unusual crack propagation in high-entropy oxide ceramics.** Oxide ceramics are always brittle with a low fracture toughness, leading to a quick propagation of cracks and premature failure of ceramic components, which becomes a major concern in engineering applications[50,51]. It is widely known that a large number of mobile dislocations are present in metals, which account for the high plastic deformation and large fracture toughness[52,53]. In contrast, in ceramics, although dislocations can be mobile in a few single-crystalline ceramics, such as $SrTiO_3$, MgO, the nucleation of dislocations is rather difficult at the crack tip due to the high bond strength of ceramics[20]. Therefore, dislocation-induced toughening is absent in normal ceramics. However, it has recently been identified that high pre-existing dislocation density can enable local plasticity on the surface of a $SrTiO_3$ crystal and may become an additional toughening mechanism in bulk ceramics[20]. Meanwhile, the stress field around the dislocations can also contribute to

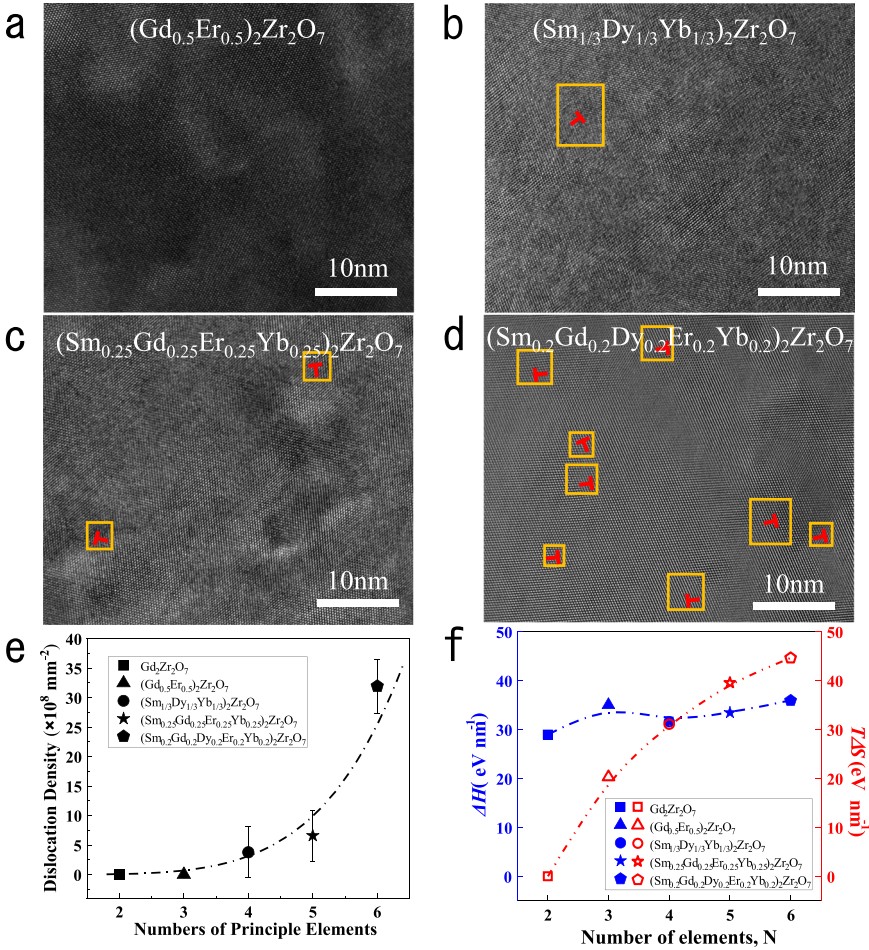

**Fig. 3 High-resolution characterization and dislocation density of samples with different principle elements. a** $(Gd_{0.5}Er_{0.5})_2Zr_2O_7$; **b** $(Sm_{1/3}Dy_{1/3}Yb_{1/3})_2Zr_2O_7$; **c** $(Sm_{0.25}Gd_{0.25}Er_{0.25}Yb_{0.25})_2Zr_2O_7$; **d** $(Sm_{0.2}Gd_{0.2}Dy_{0.2}Er_{0.2}Yb_{0.2})_2Zr_2O_7$; **e** Dislocation density varies with numbers of principle elements (The error bars represent the standard deviation of calculated dislocation density.); **f** Enthalpy change (blue line and symbols) of an edge dislocation compared with the entropy gain "$T\Delta S$" (red line and symbols) with increasing number of the principle elements. The yellow squares mark the area around the dislocations.

enhanced toughness through the shielding effect[21]. Nevertheless, both routes require high-density pre-existing dislocations ($>10^8$ mm$^{-2}$) in bulk ceramics[20,21], which is beyond the reach of the available techniques. As we have demonstrated the high-density dislocations ($10^9$ mm$^{-2}$) in oxide ceramics through the entropy gain, the mechanical properties are then studied to explore the effect of these dislocations. The crack of $Gd_2Zr_2O_7$ and HEPO ($Ca_{1.2}Gd_{0.8}Zr_{0.8}Nb_{0.6}Ta_{0.6}O_7$) fabricated by the indention method is investigated by SEM shown in Fig. 5a, b. The crack propagates transgranular in $Gd_2Zr_2O_7$ without any occurrence of intergranular fracture. While for high-entropy oxide ceramics, although transgranular fracture mode still dominates the crack propagation, some toughening mechanisms can be found in the SEM graph, in terms of intergranular fracture, crack deflection and crack bridging. To give a more objective and comprehensive view of the crack propagation behavior of HEPO, more SEM images revealing the interaction between the crack tip and dislocations are shown in Supplementary Fig. 18. Generally, the distribution of grain size could influence the cracking mode in ceramics. Therefore, the grain size distribution of $Gd_2Zr_2O_7$ and HEPO is analyzed by counting ~200 grains shown in Supplementary Fig. 17. It is confirmed that the average grain size is quite close to each other and the effect of grain size on mechanical toughness can be excluded. Due to the significant increase in dislocation density of HEPO, it is assumed that these toughening mechanisms originate from the

interaction between cracks and dislocations. The fracture toughness of HEPO measured via single edge notch beam (SENB) according to the standard ASTM C1421-18 is shown in Table 1. Compared with the fracture toughness calculated by the mixing law, the fracture toughness of HEPO ($2.46 \pm 0.32$ MPa·m$^{0.5}$) is improved by ~70%.

In order to confirm the toughening mechanism of the dislocations, the interaction between cracks and the dislocations in HE ceramics is elucidated by means of MD simulations. The HEPO model is established in Fig. 5d, in which a crack with a length of 3 nm paralleling two edge dislocations is inserted. The Burges vector of the inserted dislocation in the HEPO model is $\pm a/2[110]$. The model is relaxed at 300 K for 30 ps to eliminate the internal stress, and then deformation is applied in the y-direction with a strain rate of 1%/ps up to a maximum deformation of 30%.

Figure 5c-e displays the atomic configuration of $Gd_2Zr_2O_7$ and HEPO under various strains during the deformation process. Similar to the perfect HE model shown in Fig. 2c, a strain field originating from the two edge dislocations also exists in the HEPO model at the early stage ($\varepsilon = 0\%$). As shown in Fig. 5c-e, the compressive strain is in the red region where atoms have larger coordination numbers, and the tensile strain is in the green or blue region where atoms have smaller coordination numbers.

The simulation results (Fig. 5c and Supplementary Movie 1) of pyrochlore $Gd_2Zr_2O_7$ have good consistency with the experimental

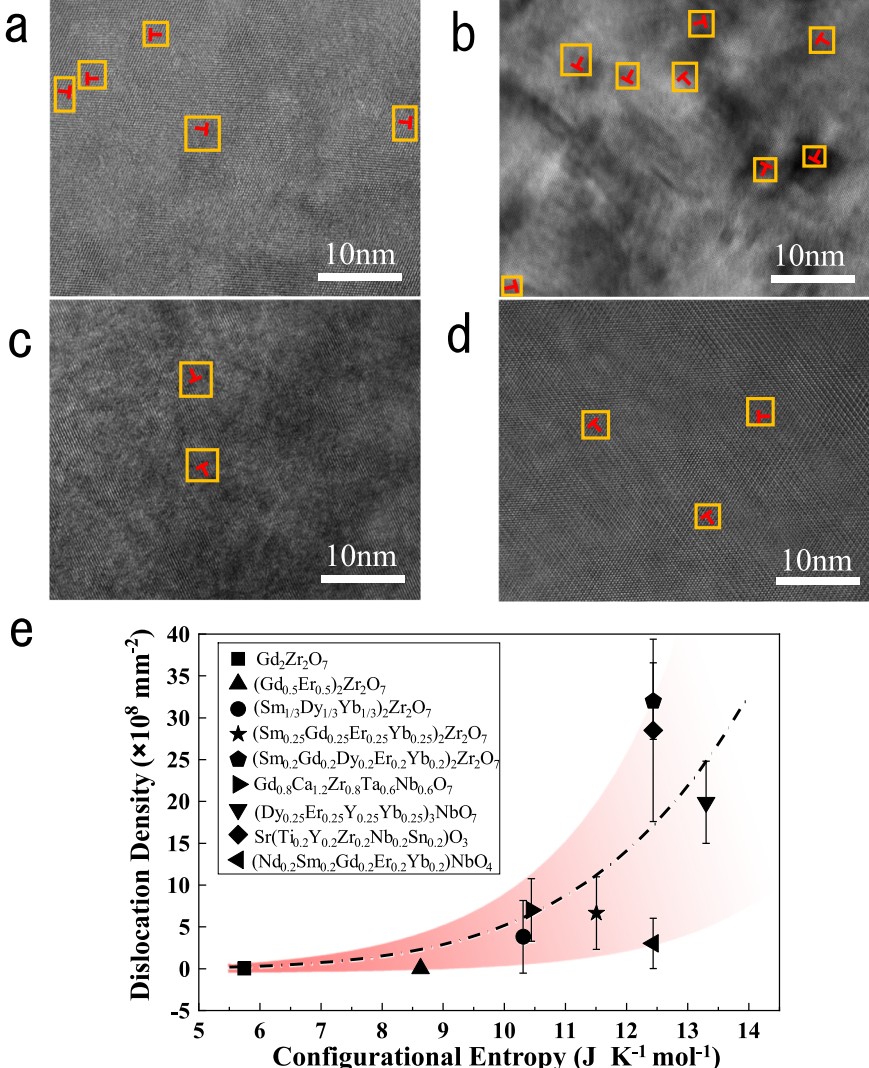

**Fig. 4 High-resolution observation of ultra-dense dislocations in high-entropy oxides with various structure. a** Fluorite oxides $(Dy_{0.25}Er_{0.25}Y_{0.25}Yb_{0.25})_3$ $NbO_7$; **b** Perovskite oxides $Sr(Ti_{0.2}Y_{0.2}Zr_{0.2}Nb_{0.2}Sn_{0.2})O_3$; **c** fergusonite oxides with monoclinic phase $(Nd_{0.2}Sm_{0.2}Gd_{0.2}Er_{0.2}Yb_{0.2})NbO_4$; **d** Pyrochlore oxides $Ca_{1.2}Gd_{0.8}Zr_{0.8}Nb_{0.6}Ta_{0.6}O_7$; **e** Dislocation density varies with configurational entropy. The error bars represent the standard deviation of calculated dislocation density. The yellow squares mark the area around the dislocations.

result (Fig. 5a), in which the crack propagates straightly without any crack deflection or bridging. For the HEPO model, the volume of crack increases with the increase of applied strain during the deformation process presented in Fig. 5d. New cracks nucleate near the two edge dislocations when the external strain reaches 5%, which takes precedence over the propagation of the inserted crack. The newly formed micro-cracks propagate and merge together with each other to form an intact crack when the external strain reaches 7% according to the Supplementary Movie 2. At the same time, the inserted crack begins to propagate in two opposite directions. The crack propagation is interrupted by the newly formed crack and then promotes a crack bridging. Moreover, the crack deflection toughening mechanism is also found via MD simulation shown in Fig. 5e and Supplementary Movie 3. Furthermore, the dislocation-induced microcrack may be different from the conventional concept of micro-cracking. For the dislocation-induced microcrack, it does not exist in the materials before meeting the macro-crack. It forms simultaneously with the macro-crack due to the superposition of the strain field of dislocations and external load. The major role is to cause the bridging of the macro-crack, as seen in the SEM image. In the

strain-stress curve of the SENB test, R-curve behavior, which is one of the important characteristics of microcrack toughening, is not observed[54]. The SENB test and MD simulation are also applied to HEFO and the results are shown in Supplementary Fig. 19 which exhibits the same toughening mechanisms. The simulation results have good agreement with the experimental results that crack deflection and bridging are present in HEFO and HEPO ceramics. Therefore, the macroscopic experimental results can be explained by the multiple interactions between cracks and microscopic ultra-dense dislocations.

Dislocation toughening also occurs in HEFO, but the effect is lower than that in HEPO as shown in the Supplementary Fig. 19. The dislocation density of HEFO is about triple that of HEPO. However, the dislocations in HEPO have a larger strain zone with a radius of 3–3.5 nm and higher strain energy, which can interact with crack tips more easily. More details can be found in Supplementary Figs. 12, 15, and 20. The larger area of strain field in HEPO $(Ca_{1.2}Gd_{0.8}Zr_{0.8}Nb_{0.6}Ta_{0.6}O_7)$ may be associated with the cations having multiple valences and a more complicated crystal structure. Therefore, there could be more chances of deflection and bridging in the HEPO due to the high energy and

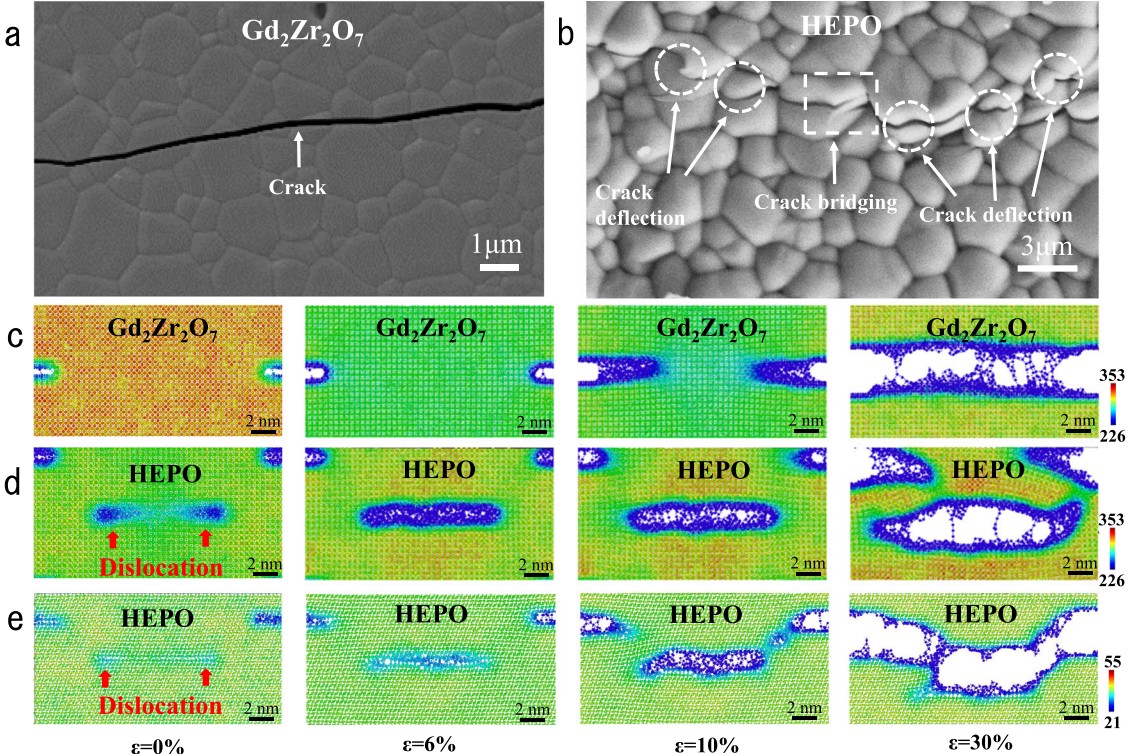

**Fig. 5 Coordination number distribution in Gd₂Zr₂O₇ and Ca₁.₂Gd₀.₈Zr₀.₈Nb₀.₆Ta₀.₆O₇ (HEPO) during crack propagation along with the experimental observations. a, b** The surface crack propagation path in $Gd_2Zr_2O_7$ and HEPO samples; **c** Straight and brittle crack propagation in $Gd_2Zr_2O_7$ crystal; **d** Crack bridging in HEPO interacting with two edge dislocations; **e** Crack deflection in HEPO interacting with two edge dislocations. Dislocations are indicated by red arrows. The color scale denotes the coordination number of each cation (unit: counts).

**Table 1 Fracture toughness of starting materials, HEPO, and calculated value of mixing law.**

| Samples | Ca₂Ta₂O₇ | Ca₂Nb₂O₇ | Gd₂Zr₂O₇ | Mixing law | HEPO |
|---|---|---|---|---|---|
| Fracture toughness (MPa m^0.5) | 0.91 ± 0.09 | 1.32 ± 0.08 | 1.86 ± 0.11 | 1.41 | 2.46 ± 0.32 |

large strain area of dislocations. Such a crack propagation path can absorb more fracture energy and further improve the fracture toughness of HEPO.

## Discussions

In conclusion, we observed the ultrahigh density of edge dislocations (~$10^9$ mm$^{-2}$) in high-entropy oxide ceramics without any specific processing for the first time. Those edge dislocations are uniformly distributed in various oxide ceramics such as fluorite $(Sm_{0.2}Gd_{0.2}Dy_{0.2}Er_{0.2}Yb_{0.2})_2Zr_2O_7$ and $(Dy_{0.25}Er_{0.25}-Y_{0.25}Yb_{0.25})_3NbO_7$, pyrochlore $Gd_{0.8}Ca_{1.2}Zr_{0.8}Ta_{0.6}Nb_{0.6}O_7$, perovskite $Sr(Ti_{0.2}Y_{0.2}Zr_{0.2}Nb_{0.2}Sn_{0.2})O_3$, and fergusonite $(Nd_{0.2}Sm_{0.2}Gd_{0.2}Er_{0.2}Yb_{0.2})NbO_4$. The dislocations originate from the composition complexity where configurational entropy gain can compensate for the large strain energy of the dislocations raised by the rigid ionic/covalent bonding in oxide ceramics. In addition, the density can even be systematically tuned by the composition complexity and the consequent configuration-entropy gain. Such large-density dislocations have unusual effects on the properties of ceramics. For example, compared with the fracture toughness calculated by the mixing law, HEPO showed an enormous increase in fracture toughness (up to 70%), due to the multiple interactions between cracks and the large strain field around the ultra-dense dislocations. We believe this work can provide a new perspective on the microstructure of

high-entropy oxide ceramics. This may deepen the understanding of the abnormal properties of high-entropy ceramics and accelerate the exploration of new properties and applications. This also provides a particular and controllable approach to establish ultrahigh-density dislocations in oxide ceramics towards unexplored phenomena and properties.

## Methods

**Sample preparation**. The high-entropy oxide ceramics were synthesized via solid-state reaction, with raw materials of $Nd_2O_3$ (≥99.99%), $Sm_2O_3$ (≥99.99%), $Gd_2O_3$ (≥99.99%), $Dy_2O_3$ (≥99.99%), $Er_2O_3$ (≥99.99%), $Yb_2O_3$ (≥99.99%), $ZrO_2$ (≥99.99%), CaO (≥99.99%), $Nb_2O_5$ (≥99.99%), $Ta_2O_5$ (≥99.99%), $Y_2O_3$ (≥99.99%), SrO (≥99.99%), $TiO_2$ (≥99.99%), and $SnO_2$ (≥99.99%). These raw powders were firstly calcined at 1000 °C for 10 h and then mixed directly in the stoichiometric ratio by ball milling, with ethanol as the mixing medium under a speed of 250 r/min for 10 h. The slurry was dried by rotary evaporation and then sintered at 1250 °C for 10 h to obtain pre-sintered powders. The pre-sintered powders were ball milled and rotary evaporated again using the same parameters above. Those dried powders were ground and sieved to get the final powders for pressing. The disk and strip samples were prepared by hydraulic pressing under 5 MPa, followed by cold isostatic pressing under 220 MPa. The as-pressed samples were stored in a drying oven for 24 h to release the internal stresses generated during pressing. Finally, those samples were pressureless sintered in air at 1500–1600 °C for 10 h and subsequently furnace cooling to obtain dense bulk samples.

**Structure and compositional characterization**. The microstructure was observed by scanning electron microscopy (SEM, Zeiss Merlin) with energy-dispersive X-ray spectroscopy (EDS, Oxford Instrument X-Max$^n$-20) for the analysis of element distribution. The disk samples were metallographically polished and then thermal

etched at 1450–1550 °C for 30 min before the SEM characterization. The atomic microstructure is observed by the high-resolution transmission electron microscope (HRTEM, JEOL JEM-2100F) and high-angle annular dark-field imaging of scanning transmission electron microscopy (HAADF-STEM, JEM-ARM300F). Geometric phase analysis (GPA) is carried out through a program compatible with Digital Micrograph.

**Fracture toughness measurement**. The fracture toughness was measured by a single edge notched beam (SENB, ASTM C 1421-18) method by the universal materials testing machine (Instron 5943). The sample dimensions of SENB test were $2 \times 4 \times 20$ mm, with a pre-made crack of about 0.5 W (less than 0.6 W). The $K_{IC}$ was then calculated by Eq. (3) and Eq. (4)[55]

$$K_{IC} = \frac{PS}{BW^{\frac{3}{2}}} f_1\left(\frac{a}{w}\right) \tag{3}$$

$$f_1\left(\frac{a}{W}\right) = 2.9\left(\frac{a}{W}\right)^{\frac{1}{2}} - 4.6\left(\frac{a}{W}\right)^{\frac{3}{2}} + 21.8\left(\frac{a}{W}\right)^{\frac{5}{2}} - 37.6\left(\frac{a}{W}\right)^{\frac{7}{2}} + 38.7\left(\frac{a}{W}\right)^{\frac{9}{2}} \tag{4}$$

where $P$ is the applied load, $B$ is the thickness of samples, $a$ is the crack length, and $W$ is the width of samples.

**Observation of surface crack propagation path**. In SEM images with crack propagation path, all the cracks were induced through Vickers indentation method with 1000gf (Buehler, Omnimet MHT). The cracks propagate under applied load and continuously grow under residual stress after unloading. The interaction between crack tip and dislocations can be elucidated by the patterns of cracks.

**Simulation details**. Classic molecular dynamic (MD) simulations were performed by using LAMMPS code[56]. The inter-ionic potential, including a short-range Born-Maryer-Buckingham (BMB) and long-range Coulombic potential, was used to describe the interaction between different ions. The rigid-ion mode with two opposite $\langle 111 \rangle \{10\text{-}1\}$ edge dislocations was created by a conjugate gradient (CG) structure optimization. The model was constructed by ATOMSK software[57]. All the figures of models with different specifications were created via OVITO software[58]. More details of the simulation were shown in Supplementary materials.

## Data availability
Source data are provided with this paper. Source data are further available from the corresponding author upon reasonable request.

## Code availability
LAMMPS Molecular Dynamics Simulator (LAMMPS) is available at https://www.lammps.org/. Open Visualization Tool (OVITO) is available at https://www.ovito.org/. The custom codes used in this work are available from the corresponding author upon reasonable request.

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

## Acknowledgements

This work was mainly funded by National Natural Science Foundation of China within grant nos. 52022042 (C.W.), 51590893 (Z.Y., Y.F., and C.W.), and 51902172 (Y.H., M.H., and P.Zh.), and National Key R&D Program of China within grant no. 2017YFA0700705 (M.H., W.P., and C.W.) and 2021YFB3702300 (Y.H., X.L., and C.W.).

## Author contributions

Y.H., P.Zh., and C.W. initiated the concepts. Y.H., P.Zh., and C.W. designed the experiments. Y.H., P.Zh., M.H., Y.L., P.Zo., L.L., Z.Y., and Y.F. conducted the experiments. X.L. performed the MD simulations. Y.H., P.Zh., X.L., Q.Z., W.P., and C.W. analyzed the data. Y.H., P.Zh., X.L., and C.W. wrote the manuscript. All the authors contributed to manuscript preparation. Y.H. and X.L. contributed equally. W.P., P.Zh., and C.W. jointly supervised this work.

## Competing interests

The authors declare no competing interests.
