## [Peer Review File · Nature Communications]

Title: Ultra-dense dislocations stabilized in high entropy oxide ceramicsREVIEWER COMMENTS

Reviewer #1 (Remarks to the Author):

This work reports a systematic analysis of the extremely high (edge) dislocation densities up to $\sim 10^9$ /mm² stabilized in high-entropy ceramics. The underlying mechanisms for the stabilization of dislocations are proposed, based on MD simulation, to be that the configurational entropy gain can compensate the large strain energy of the dislocations raised by the rigid ionic/covalent bonding. Furthermore, the fracture toughness has been evaluated by the SENB method in the ASTM standard. An increase of $\sim 70\%$ in the toughness is demonstrated, which is proposed to be induced by the crack deflection caused by micro-crack formation due to the large strain field around the dislocations. In light of the versatile functional properties based on dislocations in ceramics as well as the potential use of dislocations to toughen ceramics in recent years, this work may generate some inspiring aspects for the ceramic community.

However, there are some critical issues regarding especially the evaluation of the mechanical properties and underlying mechanisms that the reviewer deem must be clarified to strengthen this work and meet the criteria of Nat. Comm.

1) Regarding the dislocation analysis: in Fig. 1f, the strain field for the dislocation is inconsistent with the labeling of the dislocation symbol in Fig. 1c (see the right bottom dislocation). Please confirm and correct. The current version is misleading. A similar issue occurred in Fig. S12b, where the dislocation symbol is reversely labeled.

2) The testing protocol in SENB leads to samples that fracture completely in the end. However, looking at the SEM images containing the cracks, it is unclear how exactly these local regions containing such cracks are selected. From which surface on the fractured samples are these SEM images taken? This type of fractography analysis is extremely critical to underpin the fracture mechanisms, in this case, the dislocation-crack tip interaction. More details and clear schematic illustrations must be provided. The fracture toughness measurement using ASTM standard such as SENB is highly welcome and appreciated. However, regarding the crack tip-dislocation interaction analysis, the reviewer wonders why more straightforward method such as the indentation method was not used to generate cracks that can be easily compared on the surfaces of oxides with different dislocation densities. E.g., Vickers indentation method has been widely adopted for crack analysis, see G. R. Anstis et al. in Journal of the American Ceramic Society, 1981. This would also directly provide the hardness values for the strength evaluation, which are also known to be greatly impacted by dislocations as can be seen in recent dislocation studies in ceramics. In case the indentation method were to be considered, the drawbacks must also be carefully taken into consideration as discussed by Anstis et al.

Alternatively, crack tip fracture toughness using CTOD method (which has a much better controlled mechanical stability in comparison to the unstable crack propagation in large bulk SENB tests) would be most favorable to directly evaluate the dislocation effect on the toughness. For the CTOD method, the authors are suggested to refer to the comprehensive work by Fett et al., in Engineering Fracture

Mechanics 72 (2005) 6.

3) It is noticed that the grain size (Fig. 5a for Gd₂Zr₂O₇, Fig 5b for HEPO, and Fig. S17b for HEFO) varies a lot in the various samples tested. Grain size also has an effect on the fracture toughness (and strength) in ceramics. A grain size distribution analysis should be provided.

Fig. 5b shows also intergranular cracks, while Fig. 5a shows dominating intragranular cracks. Can the GB effect be completely ruled out?

Furthermore, consider the grain size change as well as the marginal increase of the fracture toughness (~70%), it is difficult to exclusively attribute the toughness increase simply to the high dislocation density, regardless of the simulation outcome.

4) Dislocation-induced microcracking has been revealed by the simulation as well. It is known that micro cracking in brittle ceramics can also lead to toughening, see A. G. Evans et al., *Acta Metall.*, 1985, 33 (8), 1525. Further discussion on this aspect would be helpful.

5) Some technical points: i) it makes no sense to the reviewer that that authors state that the fracture toughness has been increased by such an exact value of "approximately 74.2%", especially by considering the error bars in Table 1. It would be acceptable to state e.g., ~70% increase. ii) the error bar for the fracture toughness value in line 247 is missing.

Reviewer #2 (Remarks to the Author):

This manuscript reports the authors' discovery of ultra-dense dislocations existing stably in high entropy oxide ceramics, and discuss the effect of the former on the toughness of the latter. Both of the experiment and MD calculations have been carried out properly, and lots of very interesting data are presented, and generally very well discussed, based on which, the mechanisms underpinning have been clarified. The work overall is quite novel, and could provide a new approach to toughness improvement of structural ceramics, which would be very interested by the ceramics/materials community and several other related communities. The manuscript could be accepted after the minor issues listed below have been addressed properly by the authors.

1. English issues

In several instances, there exist minor errors/bad wordings/phrasing which need to be corrected or modified. Some of the most obvious ones are listed below (not a comprehensive list).

- 1) Page 1, Line 19, please change "widely present" to "commonly present".
- 2) Page 3, Line 44, please change "hardly tolerant" to "hardly tolerate".
- 3) Page 3, Line 54, please change "global materials properties" to "bulk materials properties".
- 4) Page 4, Line 66, please change "thermal dynamically" to "thermodynamically".
- 5) Page 6, Line 110, please change "correspond" to "corresponding".

- 6) Page 8, Line 178, please change "much more" to "many more".
- 7) Page 10, Line 224, please change "early failure" to "premature failure".
- 8) Page 10, Line 225-226, please change "has been a" to "is a".

2. Technical issues

1) Page 1, Line 30-31, "cracks are deflected and bridged in the complex ceramics with ~74.2% enhancement of mechanical toughness".

Please specify that such enhancement was only achieved with some of the test samples (rather than in all cases).

2) It's better to briefly compare/discuss the different effects of dislocations on toughness of a metal and a high entropy oxide ceramic.

3) I am very curious to know whether the existence of such ultra-dense dislocations in a high entropy oxide ceramic would also affect its mechanical strength. If possible, please present some mechanical strength data and compare.

4) Was it possible that some of the ultra-dense dislocations might be "newly" created due to the shear stress applied during the TEM sample polishing process?

Reviewer #3 (Remarks to the Author):

This work presented an interesting phenomenon in high entropy oxide ceramics, i.e., the apparent increase of dislocation density with the increase in the number of principle elements. While the experimental observations/analysis on dislocations in these ceramic materials were credible and convincing, the reviewer had some concerns on the rational behind the observed trend (dislocations being stabilized by increase in entropy).

The authors stated: "We demonstrate that these dislocations are thermal dynamically stabilized, because the configuration-entropy gain can compensate the large strain caused by the rigid ionic/covalent bonding in oxide ceramics." In these case, the reviewer would expect that the free energy change (ΔF) induced by a dislocation to be less (or even negative?) in systems with more principal elements. Was this the case? To support this, the changes in entropy and enthalpy due to the presence of a dislocation need to be calculated. Eq. 1 did not do this. Instead, it appeared to combine entropy and enthalpy of mixing (which were independent of any dislocations) with the strain energy of a dislocation, which in the reviewer's view did not support the author's statement.

Incidentally, the crack deflection/bridging shown in Fig. 5 appeared to more relate to grain boundaries, rather than was due to the presence of dislocations claimed by the authors.

Reviewer #1 (Remarks to the Author)

This work reports a systematic analysis of the extremely high (edge) dislocation densities up to $\sim 10^9$ /mm² stabilized in high- entropy ceramics. The underlying mechanisms for the stabilization of dislocations are proposed, based on MD simulation, to be that the configurational entropy gain can compensate the large strain energy of the dislocations raised by the rigid ionic/covalent bonding. Furthermore, the fracture toughness has been evaluated by the SENB method in the ASTM standard. An increase of $\sim 70\%$ in the toughness is demonstrated, which is proposed to be induced by the crack deflection caused by micro-crack formation due to the large strain field around the dislocations. In light of the versatile functional properties based on dislocations in ceramics as well as the potential use of dislocations to toughen ceramics in recent years, this work may generate some inspiring aspects for the ceramic community. However, there are some critical issues regarding especially the evaluation of the mechanical properties and underlying mechanisms that the reviewer deem must be clarified to strengthen this work and meet the criteria of Nat. Comm.

Response_0 Thanks for your positive and encouraging evaluations. We also appreciate the comments that can help improve the quality of the paper and have thoroughly revised the manuscript accordingly.

Comment_1: Regarding the dislocation analysis: in Fig. 1f, the strain field for the dislocation is inconsistent with the labeling of the dislocation symbol in Fig. 1c (see the right bottom dislocation). Please confirm and correct. The current version is misleading. A similar issue occurred in Fig. S12b, where the dislocation symbol is

reversely labeled.

Response_1: Thanks for pointing out the mistake. We have revised the labeling of the dislocation symbols in Fig. 1c and Fig. S8. We have also checked and confirmed the other dislocation symbols.

Comment_2: The testing protocol in SENB leads to samples that fracture completely in the end. However, looking at the SEM images containing the cracks, it is unclear how exactly these local regions containing such cracks are selected. From which surface on the fractured samples are these SEM images taken? This type of fractography analysis is extremely critical to underpin the fracture mechanisms, in this case, the dislocation-crack tip interaction. More details and clear schematic illustrations must be provided. The fracture toughness measurement using ASTM standard such as SENB is highly welcome and appreciated. However, regarding the crack tip-dislocation interaction analysis, the reviewer wonders why more straightforward method such as the indentation method was not used to generate cracks that can be easily compared on the surfaces of oxides with different dislocation densities. E.g., Vickers indentation method has been widely adopted for crack analysis, see G. R. Anstis et al. in Journal of the American Ceramic Society, 1981. This would also directly provide the hardness values for the strength evaluation, which are also known to be greatly impacted by dislocations as can be seen in recent dislocation studies in ceramics. In case the indentation method were to be considered, the drawbacks must also be carefully taken into consideration as discussed by Anstis et al. Alternatively, crack tip fracture toughness using CTOD method, which has a much better controlled

mechanical stability in comparison to the unstable crack propagation in large bulk SENB tests) would be most favorable to directly evaluate the dislocation effect on the toughness. For the CTOD method, the authors are suggested to refer to the comprehensive work by Fett et al., in Engineering Fracture Mechanics 72 {2005) &.

Response_2: Thanks for your comment. We measured the fracture toughness values through the SENB method using the ASTM standard, which gives a reliable value. Meanwhile, for evaluation of the dislocation-crack tip interaction, we did use the indentation method to observe the crack propagation as you suggest, including the crack images in Fig. 5a, 5b and S17b. As the crack could extend under the residual stress, this method may lead to an inaccurate value. Therefore, in our original submission, we used the SENB method to measure the fracture toughness values and indentation method to observe the crack propagation path and analyze the interaction between crack tip and dislocations. In this revised version, we added more SEM images to show the interaction between crack tip and dislocations using the indentation method, as shown in Fig. S19.

Revision_2: The experimental method of crack observation by the indentation method was added in Section 4 of Supplementary materials.

Comment_3: It is noticed that the grain size (Fig. 5a for $Gd_2Zr_2O_7$, Fig 5b for HEPO, and Fig. S20b for HEFO) varies a lot in the various samples tested. Grain size also has an effect on the fracture toughness (and strength) in ceramics. A grain size distribution analysis should be provided.

Fig. 5b shows also intergranular cracks, while Fig. 5a shows dominating intragranular

cracks. Can the GB effect be completely ruled out?

Furthermore, consider the grain size change as well as the marginal increase of the fracture toughness (~70%), it is difficult to exclusively attribute the toughness increase simply to the high dislocation density, regardless of the simulation outcome.

Response_3: Thanks for pointing out the issue. Fig. 5a, Fig 5b and Fig. S20b are all local images containing a few grains. Meanwhile there is also some difference in magnification, which leads to an impression that the grain size has a large difference among different compositions. Here, we compare the three compositions together with much more grains at the same magnification (Fig. S17). It can be noticed that the grain size is reasonably in accord with each other. We also attach an analysis on the grain size distribution by counting ~200 grains (Fig. S18). It also confirms that the average grain size is quite close to each other. Especially, the HEPO composition with 70% increase of toughness value has very close grain size and size distribution with those of $Gd_2Zr_2O_7$, which suggests that the grain size variation may not be a dominant factor for the improvement of toughness.

Figure S 1 SEM image of $Gd_2Zr_2O_7$, HEPO and HEFO.

Figure S 2 Grain size distribution of $Gd_2Zr_2O_7$, HEPO and HEFO.

To give a more objective and comprehensive view on the crack propagation behavior, we have done much more indentation tests to examine the crack path. We have randomly selected eight independent cracks as below (Fig. S19). It can be seen that most cracks show an intragranular cracking mode, in which the crack deflection and bridging are frequently observed and can be considered as the dominant toughening mechanism. We also notice that there are a few intergranular crack modes

in Fig. S19. However, these intergranular modes mainly occur around those micro-grains with grain size around 1 micrometer. According to the grain size distribution in Fig. S18, grains with size of 1 micrometer occupy the 10-20% of the total grains, so it is believed that intergranular mode is the minority cracking mode. The dominant toughening mechanism is the crack deflection and bridging within the intragranular mode in larger grains, while the intergranular mode around the micro-grains has a minority contribution.

Figure S 3 Intragranular deflection, bridging and divarication from different crack paths in HEPO samples.

Revision_3: The grain size distribution and more surface crack propagation path in HEPO were added in line 252-258 of article and Section 10, 11 of the Supplementary materials.

Comment_4: Dislocation-induced microcracking has been revealed by the simulation as well. It is known that micro cracking in brittle ceramics can also lead to toughening, see A. G. Evans et al., Acta Metall., 1985, 33 (8), 1525. Further discussion on this aspect would be helpful.

Response_4:

Thanks for your remind on this issue. The dislocation-induced microcrack may be different with the conventional concept of micro-cracking. As introduced in A. G. Evans et al., Acta Metall., 1985, 33 (8), 1525, the micro-cracks preexist in brittle materials due to the residual stress and form a process zone to interact with the macrocrack subject to load. One important characteristic of microcrack toughening is the R curve behavior in the strain-stress curve. In contrast, for the dislocation-induced micro-crack, it does not preexist in the materials before meeting the macro-crack under load. Instead, it is formed simultaneously with the macro-crack due to the superposition of the strain field of dislocations and the external load. The major role is to cause bridging of the macro crack, as seen in the SEM image. In the strain-stress curve of SENB test, we did not observe the R-curve behavior.

Revision_4: The comparison between dislocation-induced microcrack and conventional micro-cracking were added in line 290-296 of the article.

Comment_5: Some technical points: i) it makes no sense to the reviewer that that authors state that the fracture toughness has been increased by such an exact value of "approximately 74.2%"; especially by considering the error bars in Table 1. It would be acceptable to state e.g., ~70% increase. ii) the error bar for the fracture toughness value in line 247 is missing.

Response_5: Thanks for the comment. We have done the corrections.

Revision_5: Corresponding expression is modified in line 263 and line 264.

Reviewer #2 (Remarks to the Author)

This manuscript reports the authors' discovery of ultra-dense dislocations existing stably in high entropy oxide ceramics, and discuss the effect of the former on the toughness of the latter. Both of the experiment and MD calculations have been carried out properly, and lots of very interesting data are presented, and generally very well discussed, based on which, the mechanisms underpinning have been clarified. The work overall is quite novel, and could provide a new approach to toughness improvement of structural ceramics, which would be very interested by the ceramics/materials community and several other related communities. The manuscript could be accepted after the minor issues listed below have been addressed properly by the authors.

Response_0: Thanks for your positive evaluations.

Comment_1: English issues

In several instances, there exist minor errors/bad wordings/phrasing which need to be corrected or modified. Some of the most obvious ones are listed below (not a comprehensive list).

- 1} Page 1, Line 19, please change "widely present" to "commonly present".
- 2} Page 3, Line 44, please change "hardly tolerant" to "hardly tolerate".
- 3} Page 3, Line 54, please change "global materials properties" to "bulk materials properties".
- 4} Page 4, Line 66, please change "thermal dynamically" to "thermodynamically".
- 5} Page 6, Line 110, please change "correspond" to "corresponding".
- 5} Page 8, Line 178, please change "much more" to "many more".

7) Page 10, Line 224, please change "early failure" to "premature failure".

8) Page 10, Line 225-225, please change "has been a" to "is a".

Response_1: Thanks for pointing out the mistakes. We have corrected them and checked the overall manuscript.

Comment_2: Technical issues

Comment 2.1: Page 1, Line 30-31, "cracks are deflected and bridged in the complex ceramics with“ 74.2% enhancement of mechanical toughness".

Please specify that such enhancement was only achieved with some of the test samples (rather than in all cases).

Response 2.1: Corresponding expression is modified which specifies that 70% enhancement is achieved in pyrochlore structure ceramics with multiple valent cations.

Comment 2.2: It's better to briefly compare/discuss the different effects of dislocations on toughness of a metal and a high entropy oxide ceramic.

Response 2.2: Thanks for your suggestion, related comparison has been added in line 232-238.

Comment 2.3: I am very curious to know whether the existence of such ultra-dense dislocations in a high entropy oxide ceramic would also affect its mechanical strength. If possible, please present some mechanical strength data and compare.

Response_2.3: Thanks for the proposal. We measured the bending strength of starting materials and HEFO through three-point bending and the results are shown in Figure

R1. Compared with the strength calculated by the mixing law, the bending strength of HEFO is improved by approximately 10%. In metals, the presence of dislocations may decrease the strength due to the easy sliding process of dislocations. However, in normal ceramics, the Peierls stress, a measure of the slip resistance overcome by dislocation is always too large, so the dislocations are always immobile without significantly reducing the strength. As this issue would need more intense experimental and theoretical work and is not the major target of this paper, we may investigate and report the results in future.

Figure R1 The bending strength of starting materials and HEFO.

Comment 2.4: Was it possible that some of the ultra-dense dislocations might be "newly" created due to the shear stress applied during the TEM sample polishing process?

Response 2.4:

Thanks for pointing out the concern. In this paper, all the TEM samples experienced identical standard preparation procedure, including mechanical polishing, and ion-

beam milling. The stresses initiated by mechanical polishing can cause different types of defects to form in the material¹: dislocations, strain hardening, twinning, microfractures, fissures, and the dispersion of gaps, cavities, etc. However, the abrasion mechanism causes damage at a certain depth in the material, which is typically equal to three times the grain size of the abrasive used¹. That is, a few microns damaged layer exists on the surface of the sample polished by 1 μm diamond lapping films (DLF). However, this layer is then completely removed by the subsequent ion-beam milling process. As a result of the well-established procedures of the thinning processes, the prepared TEM specimens are in most cases with only little ion-beam induced artifacts and well representative of the material of interest.

As a supporting evidence, for the other two compositions $\text{Gd}_2\text{Zr}_2\text{O}_7$ and $(\text{Gd}_{0.5}\text{Er}_{0.5})_2\text{Zr}_2\text{O}_7$ with less principal elements, we also used the same thinning procedure and did not observe any dislocation in the TEM images. The results demonstrate that the TEM sample polishing process does not create new dislocations in our samples.

Reviewer #3 (Remarks to the Author)

This work presented an interesting phenomenon in high entropy oxide ceramics, i.e., the apparent increase of dislocation density with the increase in the number of principle elements. While the experimental observations/analysis on dislocations in these ceramic materials were credible and convincing, the reviewer had some concerns on the rational behind the observed trend (dislocations being stabilized by increase in entropy).

Response_0: Thanks for your positive comments.

Comment_1: The authors stated: "We demonstrate that these dislocations are thermal dynamically stabilized, because the configuration-entropy gain can compensate the large strain caused by the rigid ionic/covalent bonding in oxide ceramics." In these case, the reviewer would expect that the free energy change (ΔF) induced by a dislocation to be less (or even negative?) in systems with more principal elements. Was this the case? To support this, the changes in entropy and enthalpy due to the presence of a dislocation need to be calculated. Eq. 1 did not do this. Instead, it appeared to combine entropy and enthalpy of mixing (which were independent of any dislocations) with the strain energy of a dislocation, which in the reviewer's view did not support the author's statement.

Response_1:

Thanks for your comment that helps us to make this important issue clearer. To evaluate the thermodynamic stability of a dislocation, we calculate the total free energy of those atoms that comprise a dislocation (atoms confined within the dislocation width,

namely, a cylinder of atoms centered around the dislocation line with a radius around 2nm). It should be mentioned that these atoms not only have local atomic strain defined by the dislocation, but also have atomic disorder as that in high entropy materials. We had tried to make this calculation in the original Equation 1, but had some mistake.

$$\Delta F = \Delta H_{mix} + \langle E_{strain} \rangle - T\Delta S$$

The enthalpy ΔH_{mix} represents the formation energy of high entropy oxides by mixing the individual components. $\langle E_{strain} \rangle$ is the mean strain energy raised by the dislocation. In fact, the total enthalpy increase in the target atoms (atoms confined within the dislocation width) includes both contributions of ΔH_{mix} and $\langle E_{strain} \rangle$. Therefore, we use the term ΔH to represent the sum of ΔH_{mix} and $\langle E_{strain} \rangle$.

$$\Delta F = \Delta H - T\Delta S$$

For the calculation of entropy, we previously only considered the mixing entropy of the atoms due to the disorder, which caused some errors. We add the entropy increase due to the formation of a dislocation. The contribution of dislocations on the entropy increase can be determined via the definition of Shannon entropy^{2, 3}. The entropy difference ($\Delta S_{dislocation}$) can be described by the following relationship,

$$\Delta S_{dislocation} = -k_B [p_i \ln p_i + (1 - p_i) \ln(1 - p_i)]$$

Where k_B is the boltzman constant, p_i is the possibility to detect the dislocation in the model which is proportional to the total number of cations in the dislocation line (~0.34%). For the HE-Model, the calculated entropy difference is 0.0236 k_B per cation which is ~1.4% of the mixing entropy. As a result, the total entropy difference ΔS can be obtained by combining the $\Delta S_{dislocation}$ and ΔS_{mix} . Accordingly, we have updated the results in Fig 2d and Fig 3f, for which the conclusions remain unchanged.

In materials with less principal elements where the contribution of ΔS_{mix} is zero or small, the total free energy is still positive. Therefore, dislocations cannot be stabilized in those materials, which is consistent with the experimental observations. In contrast, for the high entropy compositions, the enthalpy increase can be compensated by the entropy of the dislocation and the mixing entropy of the atoms that comprise the dislocation. Therefore, we conclude that dislocation is stable by coexisting with the atomic disorder in the high entropy oxide ceramics.

Revision 2: Line 150-165 and line 169-171 have been rewritten. Data in Fig 2d and Fig 3f are updated.

Comment 2: Incidentally, the crack deflection/bridging shown in Fig. 5 appeared to more relate to grain boundaries, rather than was due to the presence of dislocations claimed by the authors.

Response 2:

We have done much more indentation tests to examine the crack propagation path and randomly selected eight independent cracks as below (Fig. S19). It can be seen that most cracks show an intragranular cracking mode, in which the crack deflection and bridging are frequently observed and can be considered as the dominant toughening mechanism. We also notice that there are a few intergranular crack modes in Fig. S19. However, these intergranular modes mainly occur around those micro-grains with grain size around 1 micrometer. According to the grain size distribution in Fig S18, grains with size of 1 micrometer occupy the 10-20% of the total grains, so it is believed that intergranular mode is the minority cracking mode. The dominant toughening

mechanism is the crack deflection and bridging within the intragranular mode, while the intergranular mode around the micro-grains has a minority contribution.

Figure S 4 Intragranular deflection, bridging and divarication from different crack paths in HEPO simples.

Revision_2: We have added more discussions in line 252-258. We also added Fig. S19 in the supplementary information.

References

1. Ayache J, Beaunier L, Boumendil J, Ehret G, Laub D. *Sample Preparation Handbook for Transmission Electron Microscopy: Techniques*. Springer New York, 2010.
2. Abzaev Y, Trishkina L, Porobova S, Klopotov AA, Vlasov VA, Klopotov V. Evolution of the Entropy of Dislocation Structures with Strain in Solid Solutions of Cu-0.5at.% Al. *Key Engineering Materials* 2016, **683**: 232-236.
3. Kohlstedt DL. 2.14 - Properties of Rocks and Minerals – Constitutive Equations, Rheological Behavior, and Viscosity of Rocks. In: Schubert G (ed). *Treatise on Geophysics*. Elsevier: Amsterdam, 2007, pp 389-417.

REVIEWERS' COMMENTS

Reviewer #1 (Remarks to the Author):

I would like to thank the authors for addressing my comments and concerns. The revised work is quite extensive and rather complete and will stir some interest in the ceramics community. I recommend the publication of this work.

Reviewer #2 (Remarks to the Author):

The authors have addressed properly all the issues raised by the three reviewers in the last round of review, and improved the manuscript accordingly. The revised manuscript is now acceptable for publish on NC, in its present form.

Reviewer #3 (Remarks to the Author):

The authors have addressed my concerns.

While the manuscript was under revision, the reviewer noticed an existing work in the literature that observed and analyzed by HRTEM the dislocations in ceramics. This work is quite relevant to the study under consideration and may be worth being cited and properly acknowledged in the manuscript.